# Peptidoglycan-tethered and free forms of the Braun lipoprotein are in dynamic equilibrium in *Escherichia coli*

**Yucheng Liang[1], Jean-Emmanuel Hugonnet[1], Filippo Rusconi[1,2]\*[†], Michel Arthur[1]\*[†]**

[1]Centre de Recherche des Cordeliers, Sorbonne Université, INSERM, Université de Paris, Paris, France; [2]GQE-Le Moulon/PA, Université Paris-Saclay, INRAE, CNRS, AgroParisTech, IDEEV, Gif-sur-Yvette, France

**\*For correspondence:**
filippo.rusconi@universite-paris-saclay.fr (FR);
michel.arthur@crc.jussieu.fr (MA)

[†]These authors contributed equally to this work

**Competing interest:** The authors declare that no competing interests exist.

**Abstract** Peptidoglycan (PG) is a giant macromolecule that completely surrounds bacterial cells and prevents lysis in hypo-osmotic environments. This net-like macromolecule is made of glycan strands linked to each other by two types of transpeptidases that form either 4→3 (PBPs) or 3→3 (LDTs) cross-links. Previously, we devised a heavy isotope-based PG full labeling method coupled to mass spectrometry to determine the mode of insertion of new subunits into the expanding PG network (Atze et al., 2022). We showed that PG polymerization operates according to different modes for the formation of the septum and of the lateral cell walls, as well as for bacterial growth in the presence or absence of β-lactams in engineered strains that can exclusively rely on LDTs for PG cross-linking when drugs are present. Here, we apply our method to the resolution of the kinetics of the reactions leading to the covalent tethering of the Braun lipoprotein (Lpp) to PG and the subsequent hydrolysis of that same covalent link. We find that Lpp and disaccharide-peptide subunits are independently incorporated into the expanding lateral cell walls. Newly synthesized septum PG appears to contain small amounts of tethered Lpp. LDTs did mediate intense shuffling of Lpp between PG stems leading to a dynamic equilibrium between the PG-tethered and free forms of Lpp.

## eLife assessment

This **useful** study describes a single set of label-chase mass spectrometry experiments to confirm the molecular function of YafK as a peptidoglycan hydrolase, and to describe the timing of its attachment to the peptidoglycan. Confirmation of the molecular function of YafK is helpful for further studies to examine the function and regulation of the outer membrane-peptidoglycan link in bacteria. The evidence supporting the molecular function of YafK and that lpp molecules are shuffled on and off the peptidoglycan is **solid**, however, some of the other data still remain **incomplete** in the revised version. The work will be of interest to researchers studying lipoproteins in gram negative bacteria.

## Introduction

The PG is an essential component of the bacterial cell envelope that counteracts the turgor pressure of the cytoplasm thereby preventing the cell from bursting and lysing in hypoosmotic environments (*Figure 1A*; *Rojas et al., 2018*). Structural studies of the PG (*Figure 1B*) revealed that this giant macromolecule consists of glycan strands linked to each other by transpeptidases belonging to the PBP and LDT families, which form 4→3 and 3→3 cross-links, respectively (*Glauner et al., 1988*). PBPs and LDTs contain different catalytic nucleophiles (Ser *versus* Cys) and use different stem peptides as

the acyl donor substrate (pentapeptide *versus* tetrapeptide) (*Figure 1C*; *Mainardi et al., 2005*). LDTs forming 3→3 cross-links are unessential enzymes (*Magnet et al., 2008*), which can participate in PG maintenance (*Morè et al., 2019*) and in β-lactam resistance acquired in vitro (*Voedts et al., 2022*). The LDT family includes additional enzymes (*Figure 1C*) that catalyze either the tethering of Lpp to PG (ErfK, Ybis, and YcfS) (*Magnet et al., 2008*) or the hydrolysis of the resulting covalent link (YafK) (*Bahadur et al., 2021*; *Winkle et al., 2021*). Lpp is the most abundant lipoprotein ($10^6$ copies per cell), about one-third of which is attached to PG. This dense connection between the PG and the outer membrane (OM) is thought to stabilize the bacterial envelope because mutants deficient in the tethering of Lpp to PG are viable but display increased vesiculation and lethal susceptibility to EDTA (*Sanders and Pavelka, 2013*). In addition, Lpp controls the distance between the inner (IM) and outer membranes, a structural feature that is critical for sensing stresses that affect the bacterial envelope (*Asmar et al., 2017*; *Mathelié-Guinlet et al., 2020*).

The enzymatic activity of YafK was established by incubating PG with purified *YafK* and detecting the products of the hydrolysis of the PG→Lpp bonds (*Bahadur et al., 2021*; *Winkle et al., 2021*). Intriguingly, the deletion of *yafK* was not associated with any decrease in the overall Lpp-to-PG tethering in the absence of stress (*Bahadur et al., 2021*). The absence of phenotype associated with the deletion of *yafK* implies, in the wild-type strain, either that the enzyme was not produced (or active) or that hydrolysis of PG→Lpp bonds by YafK is compensated by the formation of the same number of PG→Lpp bonds by LDTs. Here, we used a PG labeling method (*Atze et al., 2022*) to distinguish between these two possibilities and determine the mode of cross-linking of newly synthesized Lpp to PG.

## Results and discussion

### Time-resolved characterization of the isotopic composition of the Tri→KR muropeptide by mass spectrometry

To investigate the mode of tethering of Lpp to PG, bacteria were grown in a minimal medium containing [$^{13}$C]glucose and [$^{15}$N]ammonium chloride, thus affording fully labeled PG. At the late log phase, bacteria were transferred to a light isotope medium and the PG macromolecule was extracted from culture samples collected at various times after the medium switch. Digestion of PG with proteinases and muramidases released soluble disaccharide-peptide fragments (muropeptides) and eliminated the bulk of Lpp except for its C-terminal Arg-Lys dipeptide that remained bound to tripeptide stems (muropeptide Tri→KR in *Figure 1B*). The isotopic composition of the Tri→KR isotopologues was determined by mass spectrometry (*Figure 2*) and tandem mass spectrometry (*Table 1*). This analysis revealed that tethering of Lpp to PG did generate Tri→KR isotopologues containing light and heavy moieties at both the donor (tripeptide) and acceptor (Lpp) positions.

### Newly synthesized Lpp and stem peptides are independently incorporated into PG

The PG is polymerized from lipid II consisting of the disaccharide-pentapeptide subunit linked to the undecaprenyl lipid carrier by a pyrophosphate bond. In Gram-positive bacteria, proteins are tethered to PG by sortases that use lipid II as the acceptor of the transpeptidation reaction (*Dramsi et al., 2008*). Kinetic analysis of the appearance of isotopologues of the Tri→KR muropeptide ruled out this mode of tethering for Lpp. Indeed, this would have resulted in the linking of newly synthesized Lpp to newly synthesized stem peptide in lipid II, thus leading to the formation of Tri→KR isotopologues of the new→new type, which were only detected at late times of the kinetics, when light tetrapeptide stems became abundant in the PG (*Figure 3A* and *Supplementary file 2*). The low abundance of Tri→KR isotopologues of the new→new type at early times of the kinetics also indicates that Lpp is not preferentially incorporated at locations in which new disaccharide-peptide subunits are incorporated into the expanding PG. Thus, the tethering of Lpp to PG and the incorporation of newly synthesized subunits are spatially independent.

### Tethering of Lpp to septal PG might be limited

The mode of insertion of PG subunits into the expanding PG is different in the side walls and in the septum (*de Jonge et al., 1989*). One-at-a-time insertion of newly synthesized glycan strands into the

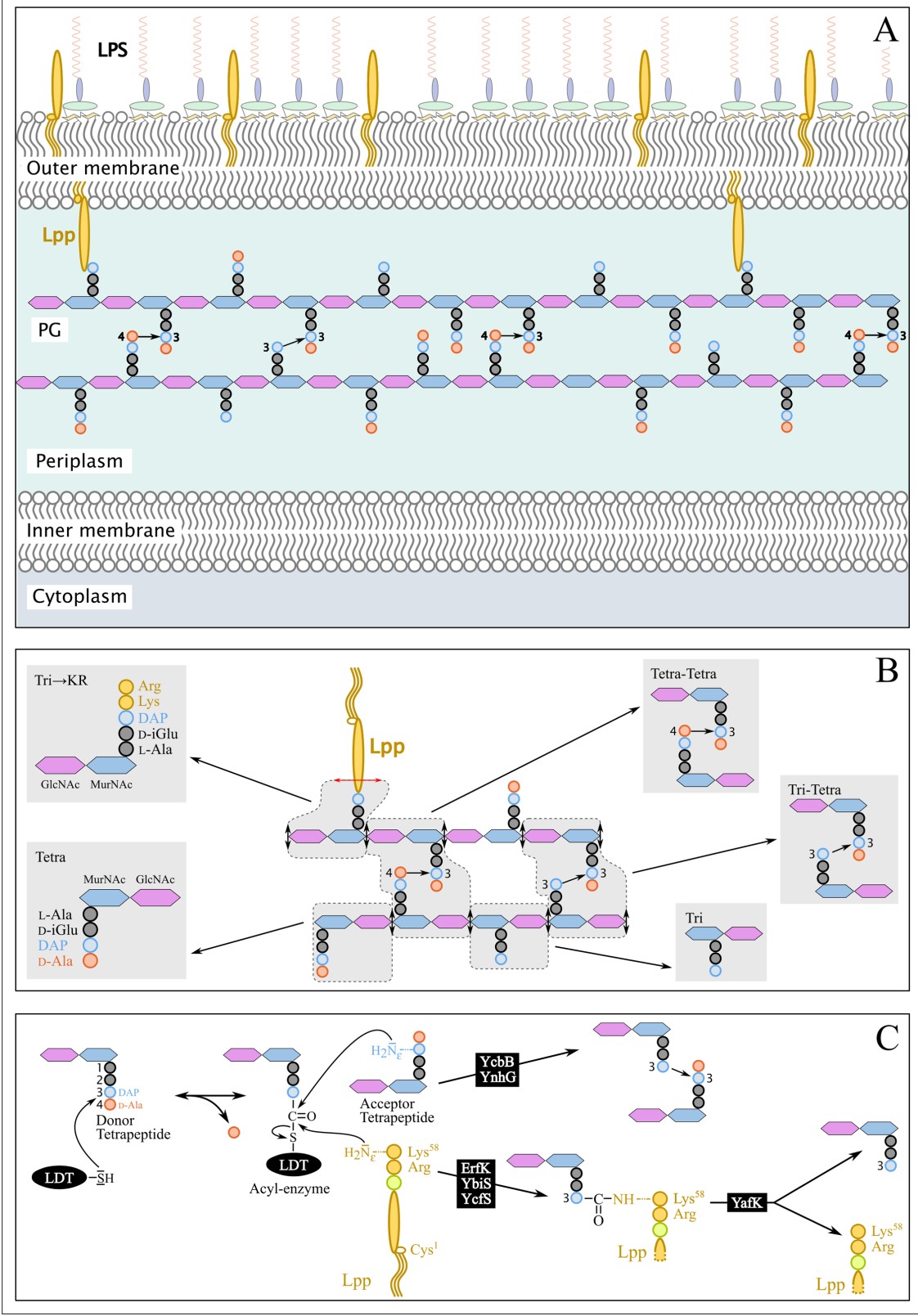

**Figure 1.** Structure and synthesis of the cell envelope of *Escherichia coli*. (**A**) Structure of the main polymers of the envelope. The envelope of gram-negative bacteria contains two membranes. The inner membrane (IM) is made of phospholipids. The outer membrane (OM) is asymmetric, with phospholipids in the inner leaflet and lipopolysaccharides (LPS) in the outer leaflet. The peptidoglycan (PG) is located in the periplasmic space delimited by the IM and the OM. The PG consists of glycan chains linked to each other by short peptides. The glycan chains consist of alternating

*Figure 1 continued on next page*

*Figure 1 continued*

β1→4-linked *N*-acetyl-glucosamine (GlcNAc) and *N*-acetyl muramic acid (MurNAc) residues. The sequence of the stem peptide linked to the D-lactoyl group of MurNAc, as assembled in the cytoplasm, is L-Ala1-D-iGlu2-DAP3-D-Ala4-D-Ala5, in which DAP is a diaminopimelyl residue. Polymerization of this disaccharide-pentapeptide subunit by glycosyltransferases results in linear glycan strands that are inserted into the expanding PG network by the combined actions of space-making hydrolases and transpeptidases. The majority of the cross-links connecting D-Ala4 of a donor stem peptide to DAP3 of an acceptor stem peptide (4→3 cross-links) are made by D,D-transpeptidases belonging to the penicillin-binding protein (PBP) family that are the essential targets of β-lactam antibiotics. L,D-transpeptidases (LDTs) catalyze the formation of 3→3 cross-links connecting two DAP residues (see panel C). The Braun lipoprotein (Lpp) provides a link between the OM and the PG since its C-terminus is covalently bound to DAP residues (panel C), whereas its N-terminal Cys residue carries three alkyl chains, one of which is carried by the amino group of the C-terminal Cys residue and the other two by a glycerol moiety linked to the sulfhydryl of that residue by a thioether bond. These alkyl chains are inserted into the inner leaflet of the OM (PG-tethered form of Lpp) or into the outer leaflet of that membrane (free form of Lpp facing the bacterial cell surface). (**B**) Determination of peptidoglycan structure. The sacculi are extracted by the hot SDS procedure, treated with pronase and trypsin (red double arrows), and digested by muramidases that cleave the MurNAc-GlcNAc bonds (black double arrows). The structure of the resulting fragments (enclosed in rounded polygons) is determined by mass spectrometry following the reduction of *N*-acetyl muramic acid (MurNAc) residues and separation by *rp*HPLC. Please note that we used the standard nomenclature for transpeptidation products in which the acyl donor and the acyl acceptor appear left and right, respectively, separated by an arrow to indicate the CO-to-NH polarity of the amide bond. (**C**) Reactions catalyzed by L,D-transpeptidases. In *E. coli*, the LDT protein family includes six paralogues that catalyze (i) the formation of 3→3 cross-links (YcbB and YnhG), (ii) the covalent anchoring of Lpp to PG (Ybis, YcfS, and ErfK), and (iii) the hydrolysis of the resulting PG→Lpp bond (YafK). Formation of the 3→3 and PG→Lpp bonds involves a common catalytic intermediate resulting from the nucleophilic attack of the DAP3-D-Ala4 bond of an acyl-donor tetrapeptide stem by the active Cys residue of the LDTs, the release of D-Ala4, and the formation of a D-Ala4-Cys thioester bond. For the formation of a 3→3 cross-link, the side-chain amino group of DAP at the third position of an acyl acceptor stem reacts with the acyl-enzyme. For the formation of a PG→Lpp bond, the side-chain of the C-terminal Lys residue of Lpp (at position 58) reacts with the acyl-enzyme.

side walls leads to a mosaic of new and old subunits gradually enriched in new subunits. In contrast, the septum mainly consists of newly synthesized PG subunits. The low abundance of new→new Tri→KR isotopologues at early time points (*Figure 3A*) thus suggests that Lpp might be incorporated in very limited amounts into nascent septal PG.

## YafK hydrolyzes tripeptide→Lpp bonds

After the medium switch, the relative abundance of old→old isotopologues is expected to decrease as a result of (i) the synthesis of new Lpp and PG stem peptides leading to a twofold decrease in one generation (*ca.* 60 min) and (ii) the hydrolysis of the tripeptide→Lpp bond by YafK. Indeed, the latter reaction releases heavy Lpp, which may remain unbound (free Lpp) or subsequently cross-linked to newly synthesized peptide stems. The decrease in the relative abundance of old→old isotopologues was slower in the absence of YafK (*Figure 3B*), thus establishing that YafK is actively hydrolyzing tripeptide→Lpp bonds.

## Hydrolysis of tripeptide→Lpp bonds by YafK contributes to the shuffling of Lpp between old and new peptidoglycan stems

At the beginning of the kinetics (0–25min), the absence of YafK had little impact on the relative abundance of old→new isotopologues (*Figure 3C*). This is expected since the bulk of the PG contains old tetrapeptide stems that act as donors in the transpeptidation reaction, thus leading to the tethering of newly synthesized Lpp to old PG. At later times, the proportion of old→new isotopologues levels off earlier in the BW25113 strain than in the Δ*yafK* mutant. This may be accounted for by the hydrolysis of old→new isotopologues by YafK, thus generating light Lpp molecules that can be cross-linked to light tetrapeptide donor stems, the abundance of which increases after the medium switch. Thus, the larger relative abundance of new→old isotopologues in the Δ*yafK* mutant provides a first indication that hydrolysis of tripeptide→Lpp bonds by YafK results in the shuffling of Lpp between PG stem peptides.

The relative abundance of new→old isotopologues was higher in the BW25113 strain than in the Δ*yafK* mutant at all incubation times (*Figure 3D*). The heavy Lpp molecules used for the formation of these isotopologues are expected to originate both from the old pool of free Lpp and from the YafK-mediated hydrolysis of old→old isotopologues. Thus, the reduction in the relative abundance of new→old isotopologues upon deletion of *yafK* additionally indicates that hydrolysis of tripeptide→Lpp bonds by YafK results in the shuffling of Lpp between PG stem peptides.

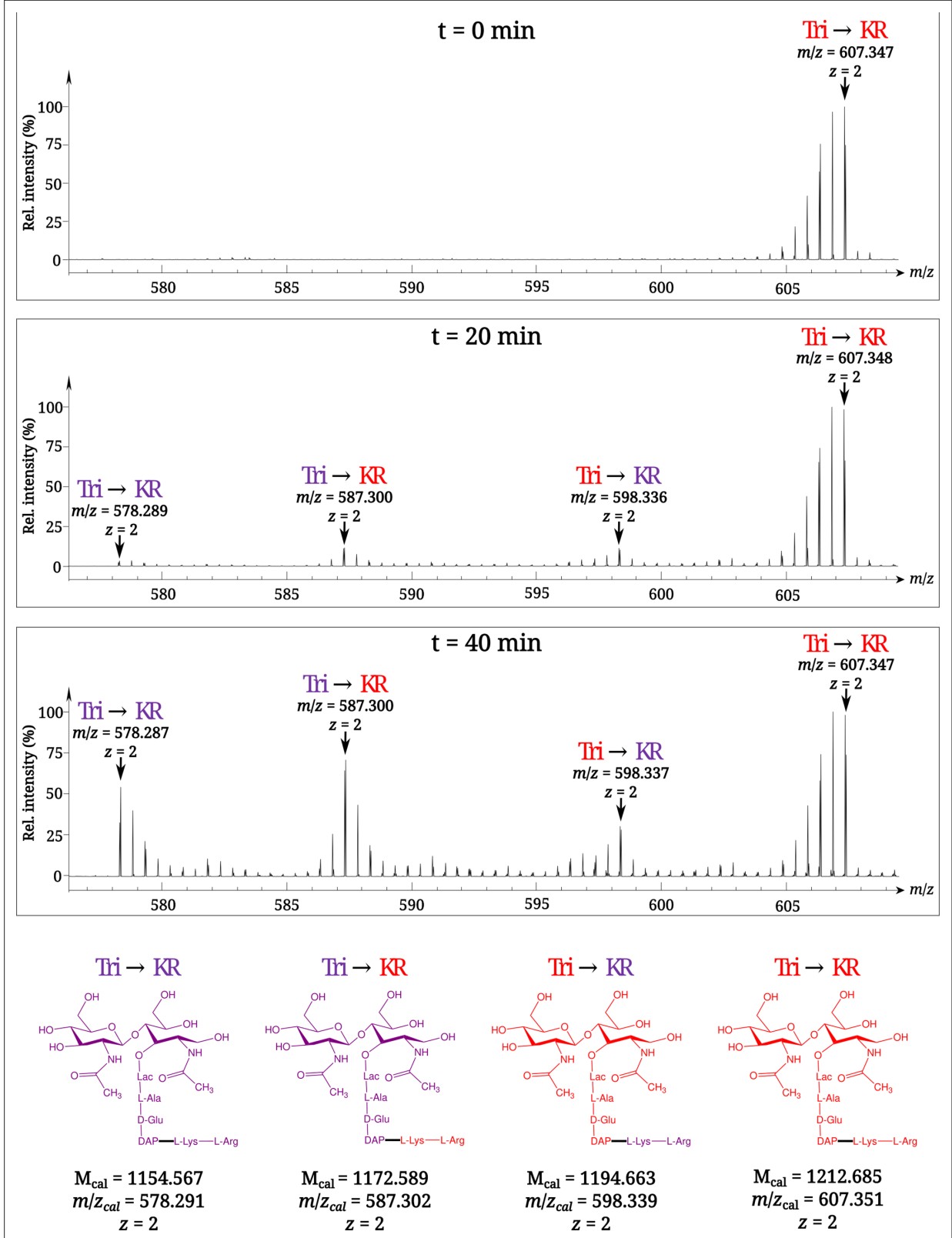

**Figure 2.** Time-resolved appearance of differentially labeled Tri→KR muropeptide isotopologues containing new (light) and old (heavy) lipoprotein (Lpp) and peptidoglycan (PG) moieties. Light and heavy residues are indicated in purple and in red, respectively. $M_{cal}$, calculated monoisotopic mass (see **Atze et al., 2022**) for the detailed description of the method used to determine $M_{cal}$ values.

**Table 1.** MS–MS analysis of the four isotopologues of the Tri⟶KR muropeptide.

| | Mass of ions generated by fragmentation of indicated isotopologues* | | | | | | | |
| | Tri⟶KR | | Tri⟶KR | | Tri⟶KR | | Tri⟶KR | |
| Deduced structure of fragments | $M_{cal}$ | $M_{obs}$ | $M_{cal}$ | $M_{obs}$ | $M_{cal}$ | $M_{obs}$ | $M_{cal}$ | $M_{obs}$ |
|---|---|---|---|---|---|---|---|---|
| GlcNAc-MurNAc$^R$-ʟ-Ala-ᴅ-iGlu-DAP⟶ʟ-Lys-ʟ-Arg$^†$ | 1,154.57 | 1,154.56 | 1,172.59 | 1,172.58 | 1,194.66 | 1,194.65 | 1,212.69 | 1,212.68 |
| MurNAc$^R$-ʟ-Ala-ᴅ-iGlu-DAP⟶ʟ-Lys-ʟ-Arg | 951.49 | 951.49 | 969.51 | 969.52 | 982.56 | 982.56 | 1,000.58 | 1,000.58 |
| ʟ-Ala-ᴅ-iGlu-DAP⟶ʟ-Lys-ʟ-Arg | 674.37 | 674.38 | 692.39 | 692.39 | 693.41 | 693.41 | 711.43 | 711.43 |
| ᴅ-iGlu-DAP⟶ʟ-Lys-ʟ-Arg | 603.33 | 603.34 | 621.36 | 621.37 | 618.37 | 618.36 | 636.39 | 636.39 |
| DAP⟶ʟ-Lys-ʟ-Arg | 474.29 | 474.29 | 492.31 | 492.31 | 483.31 | 483.31 | 501.33 | 501.33 |
| ʟ-Lys-ʟ-Arg | 302.21 | 302.21 | 320.23 | 320.23 | 302.21 | 302.21 | 320.23 | 320.23 |
| ʟ-Arg | 174.11 | 174.11 | 184.12 | 184.12 | 174.11 | 174.11 | 184.12 | 184.12 |

*Newly synthesized and existing moieties are indicated in purple and red, respectively. $M_{cal}$ and $M_{obs}$, calculated and observed monoisotopic masses (Da), respectively.
$^†$Parental ion.

## At least 50% of the tripeptide→Lpp bonds are hydrolyzed in one generation

The twofold decrease in the relative abundance of old→old isotopologues over the course of one generation, as observed in the absence of YafK (*Figure 3B*), can be fully accounted for by the synthesis of new PG stem peptides and Lpp molecules. Indeed, if the old→old Tri→KR isotopologue present at the medium shift were not hydrolyzed by YafK, its *absolute amount* would remain constant over time. However, the *relative abundance* of the old→old isotopologue decreases by 50% in one generation because the total amount of the Tri→KR muropeptide doubles in one generation (as any of the bacterial constituents). Thus, there was no detectable loss of old→old isotopologues in the Δ*yafK* strain. Furthermore, the relative abundance of old Lpp present in both the old→old and new→old isotopologues decreased twofold in 60 min both in the BW25113 strain and in the Δ*yafK* mutant (*Figure 4A and B* and *Supplementary file 3*). This implies that hydrolysis of tripeptide→Lpp bonds by YafK was not associated with any proteolytic degradation of the released Lpp. In contrast, the relative abundance of old tripeptide stems present in both the old→old and old→new isotopologues decreased more rapidly in the BW25113 strain than in the Δ*yafK* mutant (four *versus* twofold in 60 min; *Figure 4C and D*, respectively). Thus, hydrolysis of the Tri→KR bond by YafK resulted in a twofold decrease in the relative abundance of old tripeptide stems in these isotopologues, providing an estimate of 50% for the proportion of tripeptide→Lpp bonds hydrolyzed in one generation. This is a low estimate since old Lpp released by YafK can be cross-linked to a donor containing an old tetrapeptide stem, a reaction that regenerates old→old and old→new isotopologues.

## Conclusions

Here, we show that Lpp is not tethered to PG precursors prior to their polymerization, as is the case for the reaction catalyzed by sortases in Gram-positive bacteria (*Dramsi et al., 2008*). Instead, Lpp is incorporated into the expanding lateral cell walls independently from their sites of expansion. The free form of Lpp faces the bacterial surfaces raising the possibility that pools of free and PG-tethered Lpp might not be in dynamic equilibrium (*Asmar and Collet, 2018*). This is not the case because new→old isotopologues are formed in the absence of YafK (*Figure 3*), showing that existing (old) Lpp is translocated from the outer leaflet of the OM to the periplasm and tethered to neo-synthesized PG subunits. Since Lpp tethered to PG is expected to prevent OM budding, it has been proposed that vesicles are released from OM domains lacking PG-tethered Lpp (*Schwechheimer et al., 2014*; *McMillan and Kuehn, 2021*). The dynamic equilibrium between free and PG-tethered Lpp may, therefore, be physiologically relevant to spatially control the formation of OM vesicles, and thereby, their periplasmic content.

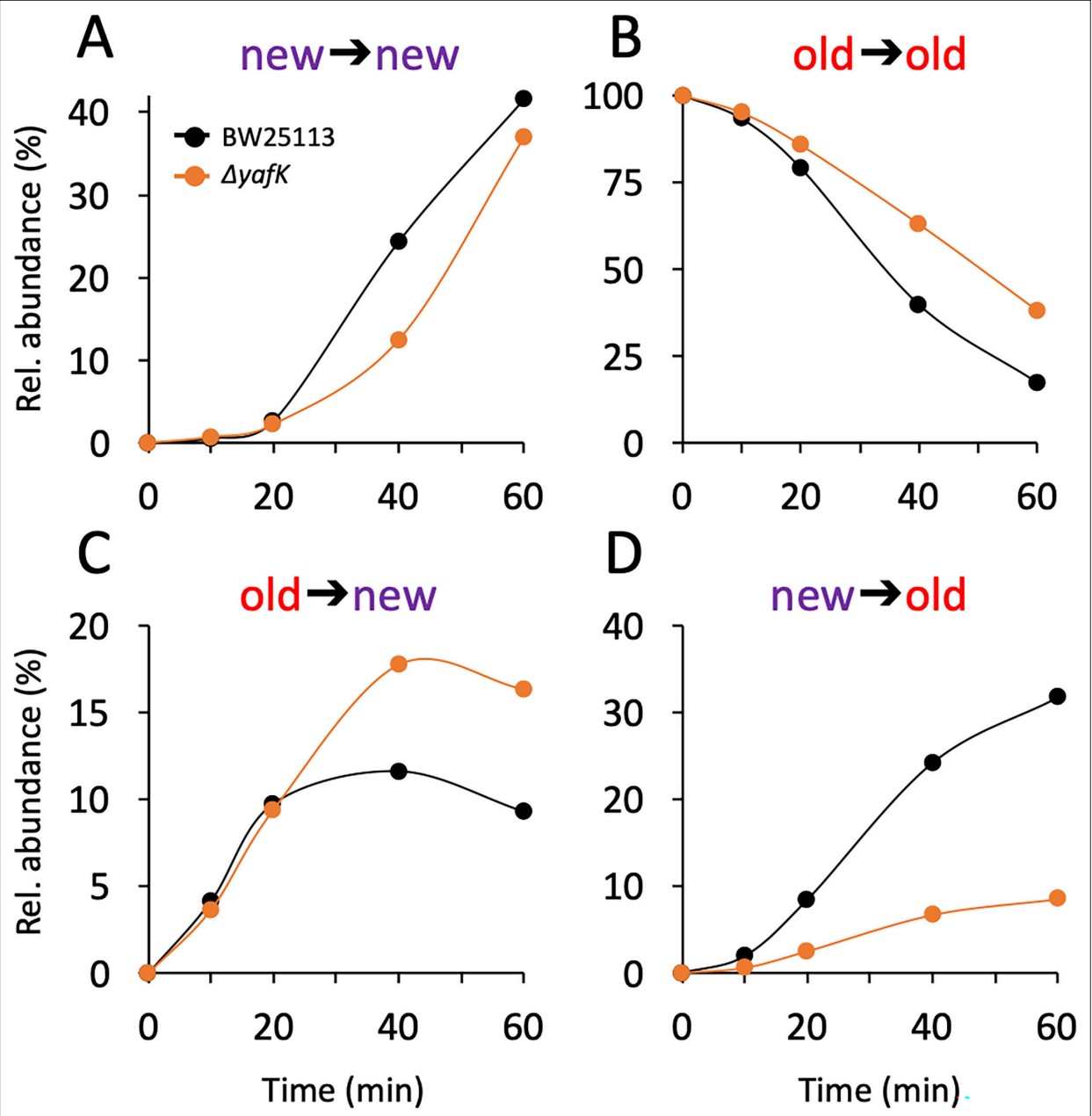

**Figure 3.** Kinetics of variations in the relative abundance of Tri→KR isotopologues. (**A**) Unlabeled isotopologue containing newly synthesized muropeptide and lipoprotein (Lpp) moieties (new→new). (**B**) Fully labeled isotopologue containing existing muropeptide and Lpp moieties (old→old). (**C**) Hybrid isotopologue containing old and new muropeptide and Lpp moieties, respectively (old→new). (**D**) Hybrid isotopologue containing new and old muropeptide and Lpp moieties, respectively (new→old). Light and heavy moieties are indicated in purple and in red, respectively. Muropeptide analysis was performed for the BW25113 strain (black) and the Δ*yafK* mutant (orange). Data are the average of three to five biological replicates. The full data set appears in ***Supplementary file 2***. Rel. abundance, relative abundance, is defined as the ratio (%) between the intensity of the indicated isotopologue and the sum of the intensities of all four isotopologues. This implies that an isotopologue that is present at the medium switch and is neither synthesized nor degraded during the following incubation of 60 min (one generation) will display a 50% reduction in its relative abundance because the total number of stem peptides doubles in one generation.

## Materials and methods

To investigate the mode of tethering of the Braun lipoprotein to PG, the *E. coli* BW25113 strain, which produces the full complement of six L,D-transpeptidases, and its derivative obtained by deletion of the *yafK* gene, were grown in minimal medium containing [$^{13}$C]glucose and [$^{15}$N]ammonium chloride to obtain fully labeled PG all procedures are described in details in ***Atze et al., 2022***. At an

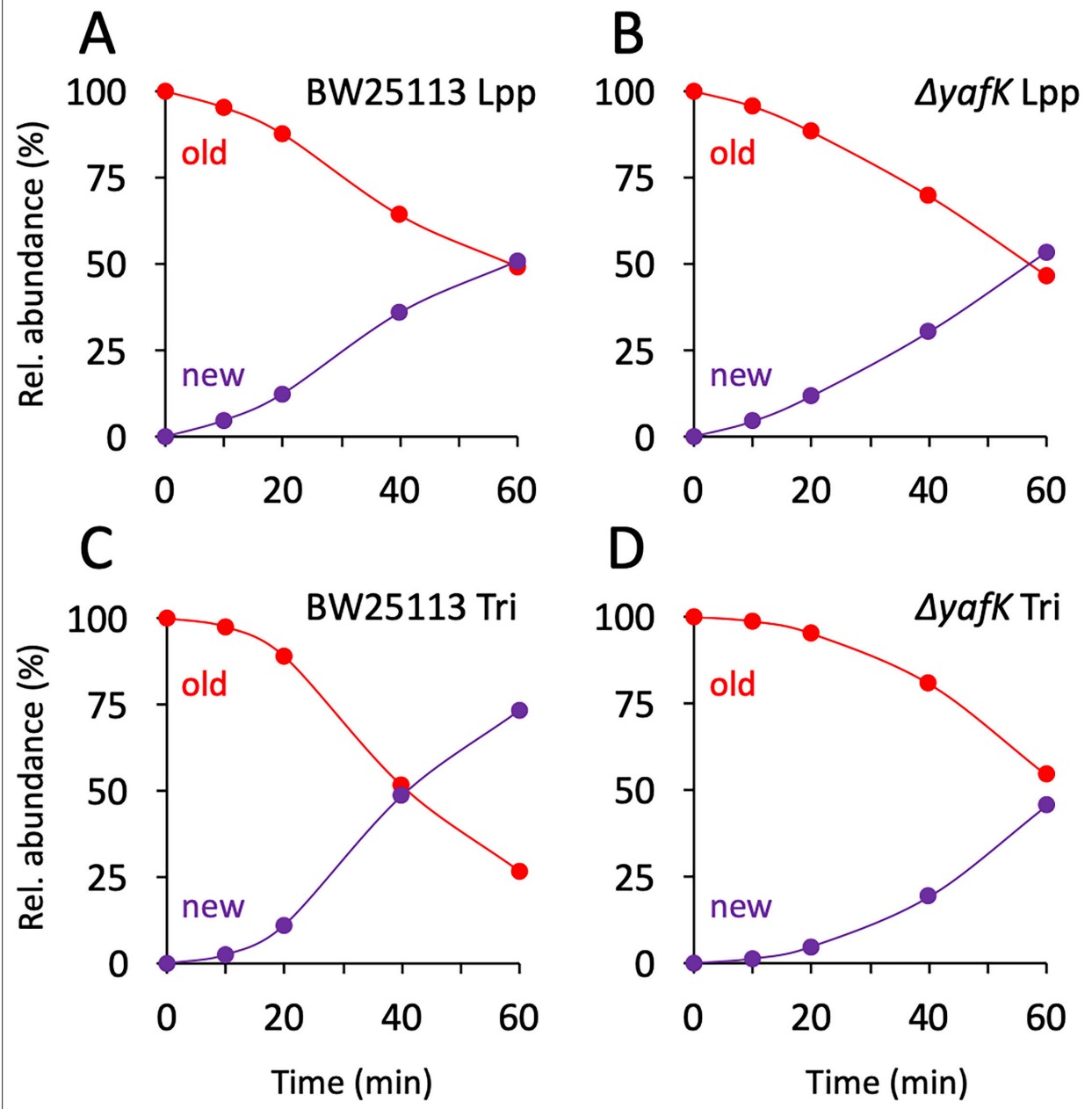

**Figure 4.** Kinetics of variations in the proportions of old and new moieties in the Tri→KR muropeptide. (**A, B**) The relative abundance of old (red) and new (purple) lipoprotein (Lpp) moieties in the isotopologues from the BW25113 strain and the Δ*yafK* mutant, respectively. The relative abundances were calculated from data appearing in *Figure 3* by adding the relative abundances of the new→old and old→old isotopologues (red curves; old Lpp) or of the new→new and old→new isotopologues (purple curves; new Lpp). (**C, D**) The relative abundance of old (red) and new (purple) tripeptide moieties in the isotopologues from the BW25113 strain and the Δ*yafK* mutant, respectively. The relative abundances were calculated from data appearing in *Figure 3* by adding the relative abundances of the old→old and old→new isotopologues (red curves; old tripeptide) or of the new→new and new→old isotopologues (purple curves; new tripeptide). Data are the average of three to five biological replicates. The full data set appears in *Supplementary file 3*.

optical density of 0.4 at 600 nm, bacteria were centrifuged, and resuspended in a minimal medium with the same chemical composition but containing the light isotopes of carbon and nitrogen ([$^{12}$C] and [$^{14}$N]) for a continued incubation of 60 min corresponding to one generation. Culture samples were collected immediately before the medium switch (time = 0) and at four times after the medium switch in order to characterize the kinetics of the replacement of heavy (existing) by light (newly synthesized) isotopes. The PG macromolecule was extracted from these culture samples by the

hot SDS procedure, digested with pronase and trypsin, and treated with muramidase's, which cleave the β-glycosidic bond connecting MurNAc to GlcNAc residues. This procedure led to the release of soluble disaccharide-peptide fragments (muropeptides) that were reduced with sodium borohydride (conversion of MurNAc to muramitol residues) and purified by rpHPLC. The isotopic compositions of the Tri→KR isotopologues was determined by mass spectrometry and tandem mass spectrometry.

## Additional information

### Funding

| Funder | Grant reference number | Author |
|---|---|---|
| Agence Nationale de la Recherche | ANR-19-CE15-0006-0 | Jean-Emmanuel Hugonnet Michel Arthur |
| National Institute of Allergy and Infectious Diseases | 1R01AI14152 | Yucheng Liang Jean-Emmanuel Hugonnet Michel Arthur |
| Agence Nationale de la Recherche | ANR-19-CE44-0007 | Jean-Emmanuel Hugonnet Michel Arthur |

The funders had no role in study design, data collection and interpretation, or the decision to submit the work for publication.

### Author contributions

Yucheng Liang, Data curation, Formal analysis, Methodology, Writing – review and editing; Jean-Emmanuel Hugonnet, Supervision, Writing – review and editing; Filippo Rusconi, Data curation, Software, Formal analysis, Methodology, Writing – original draft, Writing – review and editing; Michel Arthur, Conceptualization, Formal analysis, Supervision, Funding acquisition, Investigation, Writing – original draft, Project administration, Writing – review and editing

### Author ORCIDs

Yucheng Liang http://orcid.org/0009-0008-5939-2842
Jean-Emmanuel Hugonnet https://orcid.org/0000-0003-4150-0944
Filippo Rusconi http://orcid.org/0000-0003-1822-0397
Michel Arthur https://orcid.org/0000-0003-1007-636X

Reviewer #1 (Public review): https://doi.org/10.7554/eLife.91598.4.sa1
Author response https://doi.org/10.7554/eLife.91598.4.sa2

## Additional files

### Supplementary files

• Supplementary file 1. Time-resolved appearance of differentially labeled Tri➜KR muropeptide isotopologues containing new (light) and old (heavy) lipoprotein (Lpp) and PG moieties. This file contains mass spectral data obtained for the kinetics analyses for the BW25113 strain and the ΔyafK mutant.

• Supplementary file 2. Kinetic analysis of the relative abundance of Tri➜KR isotopologues in wild-type and ΔyafK.

• Supplementary file 3. Kinetic analysis of the content of Tri➜KR isotopologues in new and old Tri and KR moieties.

• MDAR checklist

### Data availability

The data generated in this publication (MS raw data) are available at Dryad database: https://doi.org/10.5061/dryad.t76hdr894.

The following dataset was generated:

| Author(s) | Year | Dataset title | Dataset URL | Database and Identifier |
|---|---|---|---|---|
| Hugonnet JE, Liang Y, Rusconi F, Arthur M | 2023 | Mass spectra of tri-KR muropeptide species of strains and conditions described in the paper | https://doi.org/10.5061/dryad.8467g | Dryad Digital Repository, 10.5061/dryad.t76hdr894 |

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
