## [Editor Report · eLife assessment]

This **useful** study describes a single set of label-chase mass spectrometry experiments to confirm the molecular function of YafK as a peptidoglycan hydrolase, and to describe the timing of its attachment to the peptidoglycan. Confirmation of the molecular function of YafK is helpful for further studies to examine the function and regulation of the outer membrane-peptidoglycan link in bacteria. The evidence supporting the molecular function of YafK and that lpp molecules are shuffled on and off the peptidoglycan is **solid**, however, some of the other data still remain **incomplete** in the revised version. The work will be of interest to researchers studying lipoproteins in gram negative bacteria.

---

## [Referee Report · Reviewer #1 (Public review)]

The authors present data on outer membrane vesicle (OMV) production in different mutants, but they state that this is beyond the scope of the current manuscript, which I disagree with. This data could provide valuable physiological context that is otherwise lacking. The preliminary blots suggest that YafK does not alter OMV biogenesis. I recommend repeating these blots with appropriate controls, such as blotting for proteins in the culture media, an IM protein, periplasmic protein and an OM protein to strengthen the reliability of these findings. Including this data in the manuscript, even if it does not directly support the initial hypothesis, would enhance the physiological relevance of the study. Currently, the manuscript relies completely on the experimental setup (labeling-mass spec) previously developed by the authors, which limits the broader scope and interpretability of this study.

Additionally susceptibility of strains to detergents like SDS can be tested to provide a much needed physisological context to the study.

In summary, the authors should consider revising the manuscript to improve clarity, substantiate their claims with more detailed evidence, and include additional experimental results that provide necessary physiological context to their study.

Comments on the revised version:

Regarding my comments from last review on a new figure on OMV analysis, The authors have redirected me to their previous response and have not performed the suggested control blots. I do not get their argument that this is for specialized audience. I do not have any more comments.

---

## [Author Response]

The following is the authors’ response to the previous reviews.

**Reviewer #1 (Public Review):**
The authors present data on outer membrane vesicle (OMV) production in different mutants, but they state that this is beyond the scope of the current manuscript, which I disagree with. This data could provide valuable physiological context that is otherwise lacking. The preliminary blots suggest that YafK does not alter OMV biogenesis. I recommend repeating these blots with appropriate controls, such as blotting for proteins in the culture media, an IM protein, periplasmic protein and an OM protein to strengthen the reliability of these findings. Including this data in the manuscript, even if it does not directly support the initial hypothesis, would enhance the physiological relevance of the study. Currently, the manuscript relies completely on the experimental setup (labeling-mass spec) previously developed by the authors, which limits the broader scope and interpretability of this study.

As stated in the previous response to the reviewers, MBP and RpoA were indeed used in the western blot experiments as appropriate controls for periplasmic and cytoplasmic proteins, respectively. The open review process of eLife has enabled us to include additional data from experiments suggested by the reviewers. We think that this mode of publication is appropriate in the present case for the reporting of the requested analysis of OMVs. Indeed, these data are of interest only to a rather specialized audience.

**Reviewer #2 (Public Review):**
Weaknesses:Figure 3 and 4 - why are the data shown here only two biological replicates, when there are 3-5 replicates shown in table S1 and S2? This makes it seem like you are cherry picking your favorite replicates. Please present the data as the mean of all the replicates performed, with error shown on the graph.

We apologize for forgetting to update the legend to Figures 3 and 4. In the modified version, we have indicated that the values used for the plots are the average of three to five replicates. The full set of data together with the means and standard deviations appear in Tables S1 and S2. We would like to keep the current presentation of the data because introducing standard deviations in these figures compromise the legibility of the data.

This work will have a moderate impact on the field of research in which the connections between the OM and peptidoglycan are being studied in *E. coli*. Since lpp is not widely conserved in gram negatives, the impact across species is not clear. The authors do not discuss the impact of their work in depth.

We have already answered this comment in the first response to the reviewers.